# Rotavirus infections and their genotype distribution in Rwanda before and after the introduction of rotavirus vaccination

Jean-Claude Kabayiza[1,2] , Staffan Nilsson[3] , Maria Andersson [4,5] *

1 Department of Paediatrics, School of Medicine and Pharmacy, University of Rwanda, Kigali, Rwanda,
2 Department of Paediatrics, University Teaching Hospital of Kigali, Kigali, Rwanda, 3 Department of
Laboratory Medicine, Institute of Biomedicine, Sahlgrenska Academy, University of Gothenburg, Gothenburg,
Sweden, 4 Department of Infectious Diseases, Institute of Biomedicine, Sahlgrenska Academy, University of
Gothenburg, Gothenburg, Sweden, 5 Department of Clinical Microbiology, Sahlgrenska University Hospital,
Gothenburg, Sweden

☯ These authors contributed equally to this work.
* maria.andersson.3@gu.se

pone.0284934

School of Medicine, UNITED STATES

**Data Availability Statement:** All relevant data are
within the paper and its Supporting Information
files.

## Abstract

Rotavirus vaccination has reduced mortality and hospital admissions due to rotavirus diar-
rhoea, but its effect on rotavirus infections and the impact of rotavirus genotypes are still
unclear. Real-time PCR was used to detect rotavirus and other pathogens in faeces sam-
ples from children below five years of age with acute diarrhoea, collected before (n = 827)
and after (n = 807, 92% vaccinated) the introduction of vaccination in Rwanda in 2012. Rota-
virus was genotyped by targeting VP7 to identify G1, G2, G3, G4, G9 and G12 and VP4 to
identify P[4], P[6] and P[8]. In vaccinated children, rotavirus infections were rarer (34% vs.
47%) below 12 months of age, severe dehydration was less frequent, and rotavirus was
more often found as a co-infecting agent. (79% vs 67%, p = 0.004). Norovirus genogroup II,
astrovirus, and sapovirus were significantly more often detected in vaccinated children. The
predominant rotavirus genotypes were G2P[4] and G12P[6] in 2009–2010 (50% and 12%),
G9P[8] and G1P[8] in 2011–2012 (51% and 22%), and G12P[8] in 2014–2015 (63%). Rota-
virus vaccination in Rwanda has reduced the severity of rotavirus gastroenteritis and rotavi-
rus infection frequency during the first year of life. Rotavirus infections were frequent in
vaccinated children with diarrhoea, often as co-pathogen. Rotavirus genotype changes
might be unrelated to vaccination because shifts were observed also before its introduction.

## Introduction

Acute gastroenteritis is a major cause of disease and death among young children and infants
in low-income countries. A wide range of viruses, bacteria and protozoa can induce infectious
diarrhoea. Rotavirus is one of the most important aetiologies, which prior to vaccination
caused an estimated 500,000 deaths every year. Two rotavirus vaccines, the monovalent
Rotarix (GlaxoSmithKline) and the pentavalent RotaTeq (Merck) have been available several
years, and more recently ROTAVAC (Bharat Biotech) and ROTASIIL (Serum Institute of

**Funding:** The number I previously provided was a project number (2008-333) while the number 2006-006238 is a decision number for the collaboration. The funders of this project, SIDA, is the Sweden's government agency for development cooperation. Money used in these studies has been paid in a collaboration between the University of Gothenburg and the University of Rwanda for higher education. This means that the funders had no role in study design, data collection and analysis, decision to publish, or preparation of the manuscript.

**Competing interests:** The authors have declared that no competing interests exist.

India) have been available and WHO-prequalified. Rotavirus vaccination of all children was recommended by WHO in 2009 [1], and in Africa vaccination has been implemented in the national immunization program in over 70% of countries. In Rwanda, rotavirus vaccine was implemented in May 2012 as three doses of RotaTeq vaccine given at 6, 10, and 14 weeks of age, and the vaccination coverage today reached > 97%.

Studies in Latin America, the United States and Europe have shown that rotavirus vaccination reduces severe rotavirus diarrhoea by 69–90% and hospitalization due to rotavirus by 82–98% [2–5]. A meta-analysis from studies in 24 countries between 2006 and 2016 established that vaccination reduces both mortality and severe rotavirus diarrhoea, in particular in children below 2 years of age [6]. Studies from East Africa report 40–70% reduction of hospital admissions [7–10] and a reduction (39–61%) of severe diarrhoea due to rotavirus [11–14]. A recent merge of data from the sub-Saharan Africa region shows an reduction, with internal deviations, of rotavirus positive cases from 42% during pre-vaccination period to 21% in post-vaccination period [15].

Most of the rotavirus infections in humans are caused by rotavirus genogroup A, in which several genotypes have been identified on the basis of variability in the VP7 glycoprotein (G types) and the protease-sensitive VP4 protein (P types), which surround the outer capsid. The most prevalent rotavirus genotypes, identified in more than 80% of human infections during last decades, are G1P[8], G2P[4], G3P[8], G4P[8] and G9P[8] [16]. Additionally, G12P[8] has become frequently detected in recent years [17, 18].

Rotavirus genotypes have been assessed before and after vaccine introduction in Europe, USA, Latin America and Australia [19–22]. These studies report differences in genotype distribution after vaccine introduction, which to a large extent might represent normal shifts induced by acquisition of immunity to a circulating strain, but the results from Australia suggest that the vaccination indeed may influence the spectrum of circulating genotypes. Studies on the potential impact of vaccination on circulating rotavirus genotypes in African countries are not conclusive. In some studies genotype G2P[4] and G12P[8] were frequently observed after vaccine introduction and possibly associated with hospital admission in vaccinated children [23, 24]. A report from South Africa showed a temporal association between vaccination and more complex changes in genotype distribution [25].

Polymicrobial enteric infections are frequent in low-income countries [26–28], and it is often difficult to identify the causative agent in individual cases. However, some agents, in particular rotavirus, *Shigella*, *Cryptosporidium* and *Escherichia coli* with heat stable toxin (ETEC-*estA*) have been more strongly associated with diarrhoea, and of those, rotavirus has had the highest odds ratio reflecting that it has been rarely detected among healthy controls [26–28]. If rotavirus vaccination reduces the number of rotavirus infections, or diarrhoea due to rotavirus, then one would expect, as a mathematical effect, a relative increase of the frequency of infections or diarrhoea by other diarrheagenic pathogens, but to our knowledge data on this subject are lacking.

In this work we have analysed rotavirus frequency and genotype distribution before and after the introduction of vaccine in Rwanda, and related the findings to the frequency of other enteric pathogens and to the degree of dehydration in vaccinated and unvaccinated children.

## Materials and methods

### Patients and samples

In total 1634 children with diarrhoea were included during two time periods, before and after the introduction of vaccination using the RotaTec vaccine, which according to the National Institute of Statistics of Rwanda had coverage of 98% during 2013–2015. Recruitment of

patients took place at study centres which were chosen to allow inclusion of both out-patients (health centres) or in-patients (district hospitals and university hospitals) in and around Kigali and Butare. Between November 2009 and June 2012, 827 children were recruited at five Health Centres (n = 444), three District Hospitals (n = 343) and two University Hospitals (n = 40), and between June 2014 and December 2015, 807 children were recruited at six District Hospitals (n = 736) and two University Hospitals (n = 71). The inclusion criteria during both periods were age below 5.0 years and diarrhoea, defined as passage of 3 or more loose or watery stools per day, with duration of <96 hours (with or without vomiting or fever). The samples were collected as faeces (when possible) or as rectal swabs. Demographic and clinical data (including the degree of dehydration) were recorded by a study nurse. Severe dehydration was identified according to the criteria in the WHO classification of dehydration.

## Nucleic acid extraction and real-time PCR

Faeces (approximately 250 μL) was dissolved in 4.5 mL of saline and centrifuged 5 minutes at 750x g. Then, 250 μL of the dissolved faeces or 250 μL of the rectal swab solution were mixed with 2 mL of lysis buffer, and this volume was used for extraction of total nucleic acid in an EasyMag instrument (Biomerieux, Marcy l'Étoile, France). The nucleic acids were eluted in 110 μL of which 5 μL were used for each of the 9 multiplex real-time PCR reactions targeting astrovirus, norovirus genotype I (GI) or genotype II (GII), rotavirus, sapovirus, *Campylobacter jejuni*, *Cryptosporidium parvum/hominis*, enterotoxin-producing *Escherichia coli* (ETEC) coding for heat labile toxin (*eltB*) or heat stable toxin (*estA*), enteropathogenic *Escherichia coli* (EPEC) coding for intimin (*eae*) or bundle forming pilus (*bfpA*), *Salmonella* and *Shigella*. The target genes for each pathogen and oligonucleotide sequences have been previously describe [26, 27].

Real-time PCR was performed in an ABI 7900 384-well system (Applied Biosystems, Foster City, CA) in 20 μL-reactions containing oligonucleotides and TaqMan Fast Virus 1-step Master mix (ABI, for RNA targets) or Universal Master mix (ABI, for DNA targets). After a reverse transcription step at 46˚C for 30 min followed by 10 min of denaturation at 95˚C, 45 cycles of two-step PCR was performed (15 s at 95˚C, 60 s at 56˚C). In each run, plasmids containing the target regions for all agents were amplified in parallel with patient specimens to verify the performance of each target PCR.

## Genotyping of rotavirus

All rotavirus positive samples were analysed further with a genotype specific real-time PCR targeting the VP7 (G1, G2, G3, G4, G9 and G12) and VP4 (P[4], P[6] and P[8]). This amplification was carried out in 3 parallel multiplex reactions in a Quant Studio 6 instrument (Applied Biosystems, Carlsbad, CA), as previously described [29]. Each 50 μL reaction mixture contained 10 μL of extracted sample, TaqMan Fast Virus 1-step Master mix (Applied Biosystems), and oligonucleotides specific for each genotype. After a reverse transcription step at 46˚C for 30 min followed by 10 min of denaturation at 95˚C, 45 cycles of two-step PCR was performed (15 s at 95˚C, 60 s at 58˚C) with the ramp rate adjusted to 1˚C/s.

## Sanger sequencing

If genotyping was incomplete because only either the VP7 or the VP4 genotyping PCR was reactive, Sanger sequencing of the VP7 and VP4 regions was performed as previously described [30]. Sequencing was also performed in order to verify rare genotype combination, or to identify the genotype in samples that could not be genotyped at all by the genotyping real-time PCR.

## Statistics

The frequencies of rotavirus and other pathogens between unvaccinated and vaccinated were compared, for all children and after stratification into children $\leq$ 12 months and age 12–36 months of age. To further compare the presence of pathogens other than rotavirus, the children were stratified by presence and absence of rotavirus. If vaccination protected against rotavirus infection, and if other conditions were equal, one would in children with diarrhoea expect, merely on mathematical basis that the frequencies of other pathogens would be higher in vaccinated than in unvaccinated children. If vaccination protected against rotavirus induced diarrhoea but not against rotavirus infection, one would mathematically expect higher frequencies of other pathogens in combination with rotavirus (whereas the rates in children without rotavirus should not be influenced by rotavirus vaccination).

Fisher's exact test was used to compare groups as regards categorical data and Mann-Whitney U test was used to compare groups as regards numerical data. Logistic regressions with rotavirus as outcome was used to adjust for care level (health centre, hospital) and location.

## Ethics

The study was approved by the ethical committee at the National University in Rwanda and by the regional ethical review board in Gothenburg (ID:052–08), Sweden. The study was approved by the ethical committee at the University of Rwanda and by the regional ethical review board in Gothenburg (ID:052–08), Sweden. After verbal information about the study, written informed consent was obtained from a caregiver for each child included in the study.

# Results

## Age and vaccination

Children included 2009–2012 had a mean age of 18.6 months, and those included 2014–2015, after the rotavirus vaccine introduction, a mean age of 15.9 months (14 months in vaccinated and 35.1 months in unvaccinated). Table 1 presents the number of included patients during each period, vaccinated or not vaccinated, separated in the age groups <12 months, 12–36 months and >36 months.

### Rotavirus infection rates in vaccinated and unvaccinated children

Rotavirus was detected in the same proportion of all children seeking care because of diarrhoea after (34%) as compared with before (34%) the introduction of rotavirus vaccination. There was no significant difference between vaccinated (34%) and unvaccinated (29%) children in the period after the rotavirus vaccine had been introduced either. The rotavirus frequency was however significantly lower in vaccinated (34%) than in unvaccinated (47%) children below 12 months of age (Table 2).

**Table 1. Age distribution for children that had or had not received rotavirus vaccination.**

|  |  | <12 months | 12–36 months | ≥36 months |
|---|---|---|---|---|
| Total | n = 1634 | n = 679 | n = 805 | n = 150 |
| Not vaccinated | n = 901(55%) | 330 (49%) | 430 (53%) | 141 (94%) |
| *Period 1* | n = 827 (48%) | 326 (48%) | 396 (49%) | 105 (70%) |
| *Period 2* | n = 74 (7%) | 4 (0.6%) | 34 (4%) | 36 (24%) |
| Vaccinated (*Period 2*) | n = 733 (45%) | 349 (51%) | 375 (47%) | 9 (6%) |

Period 1, November 2009 to June 2012. Period 2, June 2014 to December 2015.

**Table 2. The frequency of rotavirus infection in vaccinated and unvaccinated children by age group.**

|  | Unvaccinated | Vaccinated | OR | P | ORadj | Padj |
|---|---|---|---|---|---|---|
| Age<12 | 330 | 349 |  |  |  |  |
| Rotavirus | 155 (47%) | 117 (34%) | 0.57 | 0.0004 | 0.87 | 0.003 |
| Age 12–36 | 430 | 375 |  |  |  |  |
| Rotavirus | 128 (30%) | 133 (35%) | 1.30 | 0.1 | 0.99 | 0.93 |

OR, odds ratio; ORadj and Padj, adjusted for health care level and location.

Severe dehydration was more common in unvaccinated children who had as compared with those who did not have rotavirus infection (18.3% vs. 8.1%, p<0.01). By contrast, in vaccinated children, severe dehydration was observed at similar rates in rotavirus positive (3.6%) and rotavirus negative (4.1%) children (Table 3). The rotavirus concentration in faeces was not influenced by vaccination: the median threshold cycle value for rotavirus was identical (21.6) in unvaccinated and vaccinated children.

## Infections with other pathogens

As shown in Fig 1A and 1B and S1 Table, astrovirus, norovirus GII and sapovirus infections were significantly more common, and *Cryptosporidium* infections less common, in children that were rotavirus vaccinated than in those who were not, in particular in the group less than 12 month. This relative increase of some viral infections in vaccinated children was mainly seen in children below 12 months of age co-infected with rotavirus, but to some extent seen also in those that did not also have rotavirus.

## Co-infections between rotavirus and other pathogens

Rotavirus was significantly more often detected together with co-infecting agents (79% vs. 67%, p = 0.003), and the mean number of co-infecting pathogens was higher (mean 1.75 vs. 1.40, p<0.0001), in vaccinated as compared with unvaccinated children (Fig 2). When only unvaccinated children with rotavirus infections were compared, a smaller (non-significant) difference in the frequency and number of co-infecting pathogens after as compared with before the introduction of vaccination was seen (73% vs. 67%; mean 1.47 vs. 1.39, p = 0.55).

Severe dehydration tended to be more frequent when rotavirus was present alone as compared with as co-pathogen (24/151 [16%] vs. 40/401 [9.1%]; OR = 1.71, p = 0.07), and this

**Table 3. Degree of dehydration in children with diarrhea who had or did not have rotavirus detected in faeces, and who had or had not been rotavirus vaccinated.**

|  | Rotavirus + | | | | Rotavirus – | | | |
|---|---|---|---|---|---|---|---|---|
| **All** | **Vaccinated** | **Not vaccinated** | OR | p | **Vaccinated** | **Not vaccinated** | OR | p |
|  | **n = 251** | **n = 301** |  |  | **n = 482** | **n = 600** |  |  |
| Severe Dehydration (n = 133) | 9 (3.6%) | 55 (18.3%) | 0.17 | <0.0001 | 20 (4.1%) | 49 (8.2%) | 0.49 | 0.0082 |
| No Severe Dehydration (n = 1501) | 242 (96.4%) | 246 (81.7%) |  |  | 462 (95.9%) | 551 (91.8%) |  |  |
| <12 month | Vaccinated | Not vaccinated | OR | p | Vaccinated | Not vaccinated | OR | p |
|  | n = 117 | n = 155 |  |  | n = 232 | n = 175 |  |  |
| Severe Dehydration (n = 66) | 5 (4.3%) | 34 (22%) | 0.16 | <0.0001 | 6 (2.6%) | 21(12%) | 0.19 | 0.0002 |
| No Severe Dehydration (n = 613) | 112 (95.7%) | 121 (78%) |  |  | 226 (97.4) | 154 (88%) |  |  |
| 12–36 month | 'Vaccinated | Not vaccinated | OR | P | Vaccinated | Not vaccinated | OR | p |
|  | n = 133 | n = 127 |  |  | n = 242 | n = 301 |  |  |
| Severe Dehydration (n = 63) | 4 (3%) | 21 (16.5%) | 0,16 | 0.0002 | 14 (5.8%) | 24 (8%) | 0.71 | 0.3980 |
| No Severe Dehydration (n = 740) | 129 (97%) | 106 (83.4%) |  |  | 228 (94.2%) | 277 (92%) |  |  |

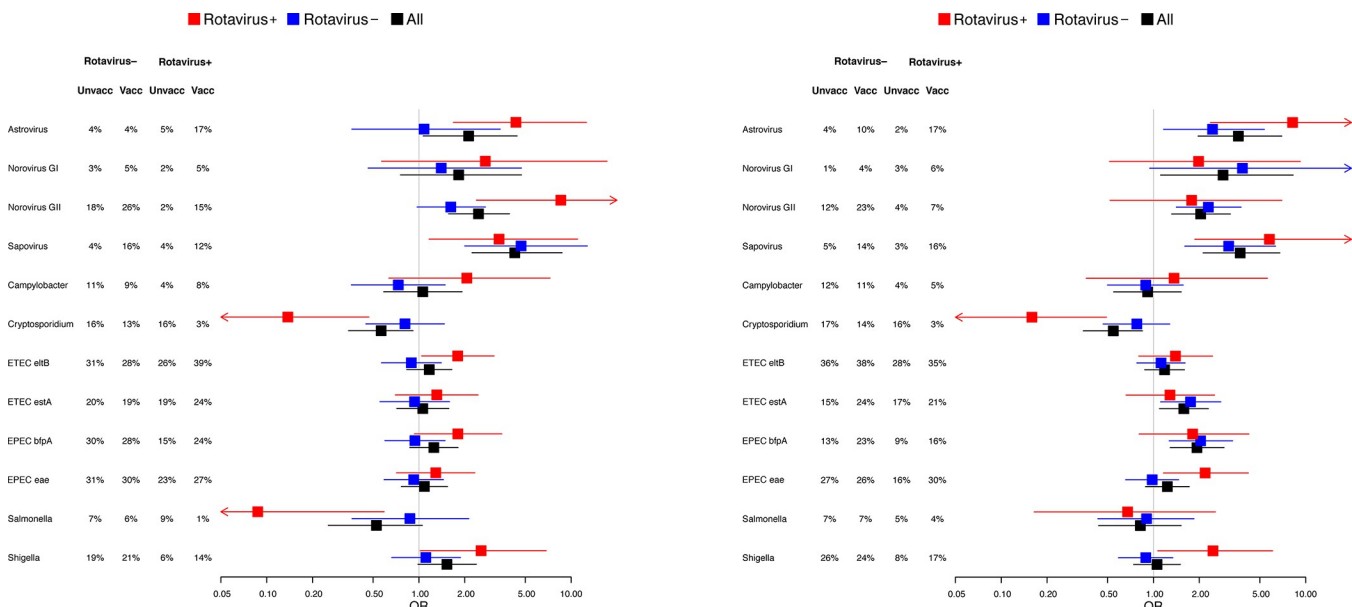

**Fig 1. The frequency of other pathogens in vaccinated and unvaccinated children shown as odds ratios (OR) and their 95% confidence intervals, for all children (black), and separated in children with (red) or without (blue) rotavirus infection. 1A** represent children below 12 months of age and **1B** children between 12 and 36 months of age.

trend was seen both in vaccinated (3/53 [5.7%] vs. 6/198 [3.0%]; OR = 1.92, p = 0.40) and unvaccinated children (21/98 [21%] vs. 34/203 [17%]; OR = 1.35, p = 0.34).

## Genotype distribution of rotavirus before and after vaccine introduction

Out of all 552 rotavirus positive samples, 505 could be genotyped (91% and 92% of the rotavirus positive samples from the periods before and after vaccine introduction). As shown in Fig 3, the genotype distribution changed considerably over time. The most common genotypes were G2P[4] (46%) during 2009–2010, G9P[8] (50%) during 2011–2012, and G12P[8] (58%) during 2014–2015, after the rotavirus vaccination was introduced. The prevalence of minor genotypes also fluctuated: G12P[6] was present only before introduction of the vaccine in 2009–2012 (12%), G1P[8] was relatively frequent in 2011 (29%) and 2015 (26%) but rare the other years, and G4P[8] and G8P[4] were present in 2014 (15% and 3%), but essentially absent the other years.

Rare genotypes, genotype mixtures, or only either a G or a P type, were observed in 4–14% of the samples over the years. Among the very rare genotypes, G4P[4], G4P[6], G8P[8], G9P[6], G12P[4] were detected in one case each, and G8P[6] in two cases during the whole study period. In total, there were 23 mixed infections with several G and/or P types present in the same sample, and in 27 samples only a G or a P type was detected. In general, detection of only either P or G, as well as the failure to detect any type in 45 samples, was explained by a low viral load, but there were samples from 2011 and 2014, in which a G type was not identified despite detection of P types (6 P[6] and 4 P[4]) with relatively low Ct values.

## Discussion

Rotavirus vaccination was introduced in Rwanda in 2012. This study shows that among children seeking care because of diarrhoea, rotavirus vaccination has reduced the frequency of rotavirus infections in those less than 12 months of age, but not in the older children.

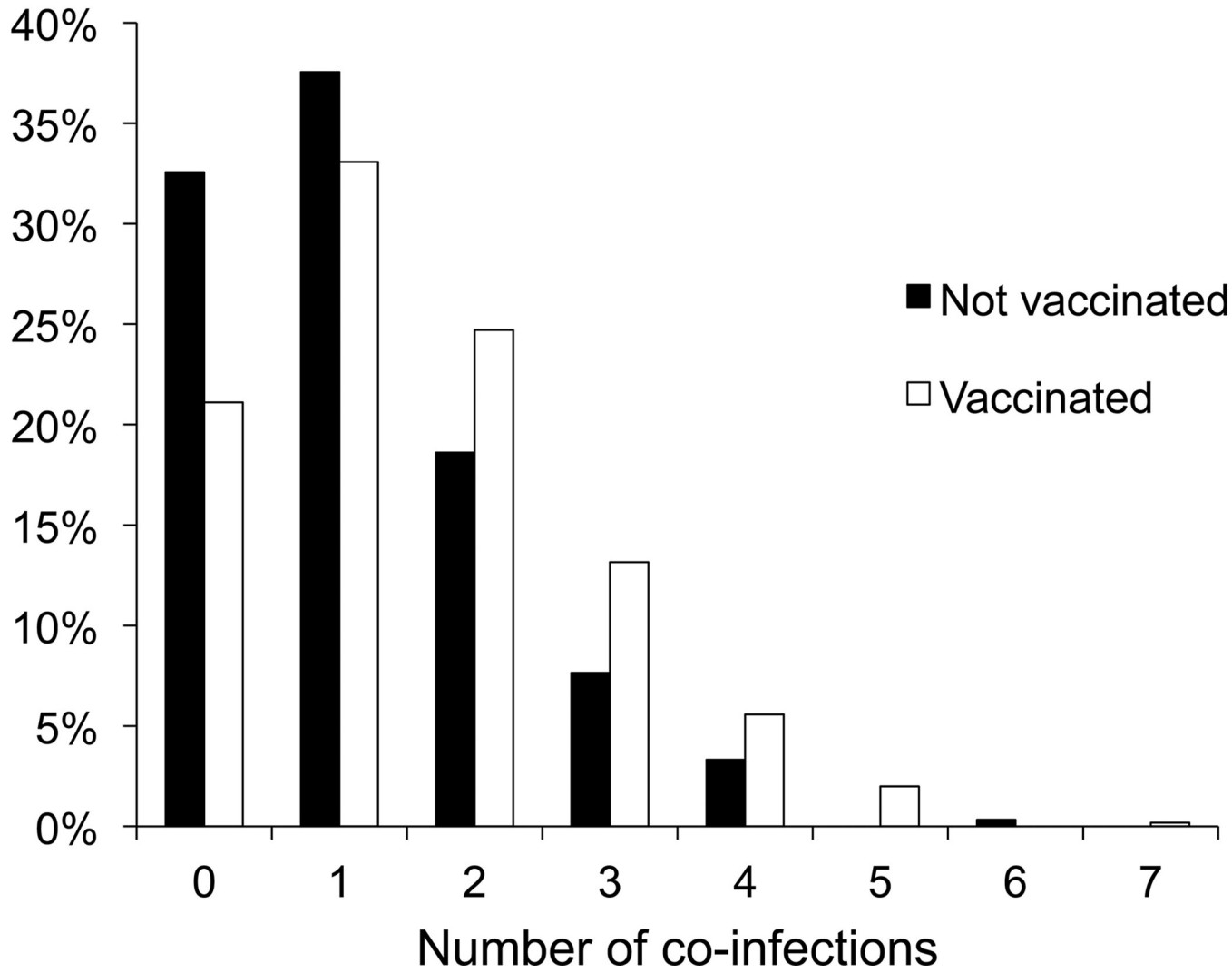

**Fig 2. The percentage of samples without or with one or several pathogens detected together with rotavirus before (2009–2010) and after (2014–2015) the introduction of rotavirus vaccination.**

Moreover, in vaccinated children, those with rotavirus infection less often had severe dehydration and rotavirus was more often presented as a co-infection together with other pathogens than in unvaccinated children.

The finding that the frequency of rotavirus infections was not generally lower in vaccinated children agrees with a recent study in Malawi in which rotavirus was detected in 32% of children with diarrhoea before and in 29% after vaccination started [8]. These rates are higher than seen in Latin America, where rotavirus has been detected in 11–20% of children with diarrheal infections after vaccination commenced [5]. High rotavirus rates subsequent years after the introduction of vaccination have also been observed in Kenya (31.5%), Ghana (26%) and Togo (36%), but in the latter two studies even higher frequencies (50% and 53%) were observed before vaccination [31–33]. Despite these reports of a prevailing high prevalence of rotavirus infections shortly after vaccine introduction several African studies have shown high vaccine efficiency against hospital admission (54–75%) [7, 32, 34–36]. In the present study, the impact on hospitalization could not be evaluated, but severe dehydration was rarer in vaccinated than in unvaccinated children also when age differences were considered.

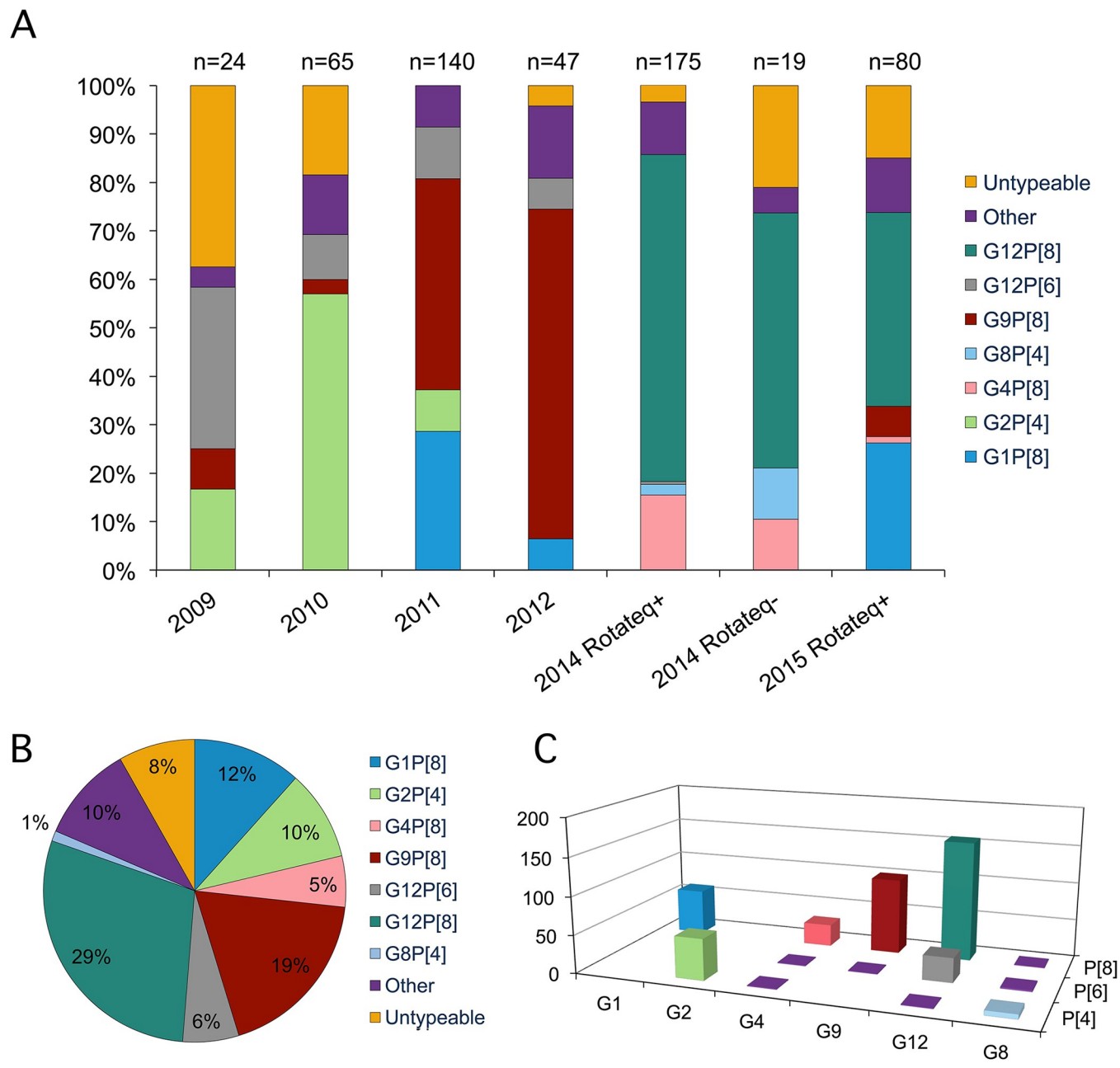

**Fig 3. A.** Genotype distribution before (2009–2010) and after (2014–2015) the introduction of rotavirus vaccination. RotaTeq+, vaccinated; RotaTeq–, not vaccinated. **B.** Rotavirus genotype distribution in all samples taken between 2009 and 2015. **C.** Number of samples with different P (VP4) and G (VP7) type combinations.

Our results show that in vaccinated children, other pathogens have become relatively more frequent, in particular norovirus GII (18% vs. 10%), astrovirus (10% vs. 4%) and sapovirus (13% vs. 4%). Similarly, previous studies have reported higher rates of norovirus GII infections in children with acute gastroenteritis after rotavirus vaccine implementation [37–39]. These increased rates in vaccinated children likely reflect a relative increase due to a decline of rotavirus diarrhoea rather than an absolute increase in number of infections. *Cryptosporidium* was less frequent after as compared with before the introduction of vaccination, possibly reflecting seasonal variation of this pathogen.

In our previous studies, rotavirus infections were rare among healthy controls, and the association with diarrhoea was stronger than for other pathogens, suggesting that rotavirus was usually the cause of diarrhoea also when other pathogens were present [26, 27]. In the present study, rotavirus was more often detected as a co-infection together with other pathogens in vaccinated as compared with unvaccinated children. In many of these cases, rotavirus was probably not the cause of diarrhoea, or at least not the main cause. The finding that rotavirus often is a co-infecting pathogen in vaccinated children with diarrhoea is an important observation of our study. It points at a risk, that the effect of rotavirus vaccination may be underestimated if other pathogens are not analysed in studies that evaluate the effect of vaccination. A resent meta-analysis of enteric pathogens among children under 5 years in sub-Saharan Africa highlighted the importance of analysing a wide range of potentially causative agents, in order to identify the true aetiologies of diarrheal disease and plan for prevention and treatment [40]. To investigate to what extent rotavirus infections in vaccinated children contribute to symptoms, additional studies including also healthy controls are required.

Even if severe disease was rare in vaccinated children, the persistent circulation of rotavirus in the population is problematic. First, it remains a threat to children who respond poorly to vaccination or who are not vaccinated. Second, circulating rotaviruses might evolve, by reassortment with each other or with animal strains, into novel and more virulent strains. Therefore, surveillance of rotavirus infections and their genotype distribution should continue, to be able to detect possible vaccine breakthroughs. A potential advantage of a continuing circulation of rotavirus might be that vaccinated persons would be boosted, and therefore maintain protective immunity against rotavirus disease.

In the present study, the spectrum of rotavirus genotype changed radically between 2009 and 2015. After rotavirus vaccination was introduced in 2012, there was a shift from predominance of G9P[8] in 2011–2012 to predominance of G12[P8] in 2014–2015 paralleled by an emergence of G4[P8] and G8[P4] and re-emergence of G1[P8]. These changes might be related to genotype differences in the protection of the RotaTeq vaccine, but it is likely that they to a large extent reflect natural genotype shifts, which have been observed in several previous studies and also were seen in the present study during the pre-vaccination period.

In summary, we report that after the introduction of rotavirus vaccination the frequency of rotavirus infections in Rwanda overall has remained high, but has decreased significantly in children less than 12 month. In vaccinated children, rotavirus infections were more rarely presented with severe dehydration and were more often accompanied by another pathogens, which likely was the cause of diarrhoea in many of these cases.

## Supporting information

**S1 Table. Pathogen detection rates in children who had or had not received rotavirus vaccine, and did or did not have rotavirus infection.**
(DOCX)

## Acknowledgments

We thank the nurses and laboratory personnel in Rwanda for their dedicated work with inclusion of patients and collection of samples in this project.

## Author Contributions

**Conceptualization:** Jean-Claude Kabayiza, Staffan Nilsson, Maria Andersson.

**Data curation:** Jean-Claude Kabayiza, Maria Andersson.

**Formal analysis:** Maria Andersson.

**Methodology:** Maria Andersson.

**Project administration:** Jean-Claude Kabayiza.

**Software:** Staffan Nilsson, Maria Andersson.

**Visualization:** Staffan Nilsson, Maria Andersson.

**Writing – original draft:** Jean-Claude Kabayiza, Staffan Nilsson, Maria Andersson.

**Writing – review & editing:** Jean-Claude Kabayiza, Staffan Nilsson, Maria Andersson.

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
