## [Decision Letter · Decision Letter 0]

26 Jan 2023

PONE-D-22-34328Rotavirus infections and their genotype distribution in Rwanda before and after the introduction of rotavirus vaccinationPLOS ONE

Dear Dr. Andersson,

Thank you for submitting your manuscript to PLOS ONE. After careful consideration, we feel that it has merit but does not fully meet PLOS ONE’s publication criteria as it currently stands. Therefore, we invite you to submit a revised version of the manuscript that addresses the points raised during the review process.

We look forward to receiving your revised manuscript.

Kind regards,

Mrinmoy Sanyal, PhD

Academic Editor

PLOS ONE

Journal Requirements:

Reviewers' comments:

Reviewer's Responses to Questions

**Comments to the Author**

1. Is the manuscript technically sound, and do the data support the conclusions?

Reviewer #1: No

Reviewer #2: Partly

2. Has the statistical analysis been performed appropriately and rigorously? 

Reviewer #1: No

Reviewer #2: Yes

3. Have the authors made all data underlying the findings in their manuscript fully available?

Reviewer #1: Yes

Reviewer #2: Yes

4. Is the manuscript presented in an intelligible fashion and written in standard English?

Reviewer #1: Yes

Reviewer #2: No

5. Review Comments to the Author

Reviewer #1: This is an interesting study with value to add to current literature. The study design is technically sound. My major concern is that causative conclusions are drawn without robust statistical analysis to support them. There is insufficient discussion of the potential limitations of the study, particularly concerning is the absence of discussion of confounding factors that may have contributed to the reduction in frequency of rotavirus infections. Further details on sampling and recruitment methods would also be beneficial - was a sample size calculation performed? Overall, there are some errors in spelling (importantly faces -> faeces/feces) and grammar that should be addressed to make the manuscript clearer.

Reviewer #2: The authors used a case-control study design to indirectly evaluate rotavirus vaccine efficacy in hospital admitted diarrhea patients in Rwanda. They compared rotavirus and other enteric pathogen detection rates in patient samples collected before and after vaccination campaign with Rotataq. The hypothesis is that if the rotavirus vaccine is effective relative proportions of rotavirus infection should be lower and other enteric pathogens should be higher in samples collected after vaccination campaign. The limitation of the paper is that because of the case-control design nature, efficacy was measured as odds ratio while number of previously published studies, including prospective studies in countries of similar economic status demonstrated vaccine protective rates. This study does not offer any significant novel findings. The other objective of the study is to investigate the rotavirus serotype shifts after vaccination with Rotataq. However, given the short period of observation and small sample size, the authors are unlikely to distinguish if the viral serotype changes are the result of vaccination or natural variation. There are other issues the authors need to address.

1. The authors used pre and post vaccination hospital admitted diarrhea patients to compare the RV infection rates in several age groups. There was not significant protection in age 12-36 month group and the overall odds ratio was moderate in age <12 month (table 2). Their results are consistent with previously reported moderate efficacy of rotavirus vaccine in low income developing countries including African countries. Because of the differences in age groups, the authors should provide more details of Rwanda rotavirus vaccine program. For instance, the vaccination starting age and how many boost immunizations and the vaccination rate in different age groups. The author should discuss if the different vaccine protective efficiency in the two age groups are the results of vaccination schedule or vaccination rate.

2. Table 1, 2, it appears that group of >36 month old had no specific values in the study. Authors should consider to remove this age group from the analysis.

3. Table S1, authors suggested that the rotavirus vaccine worked better to prevent severe dehydrating diarrhea. Since it is one of their major conclusions, the table S1 should be included in main text. Odds ratios for severe diarrhea in rotavirus positive and negative groups should be separately calculated and age difference in vaccine effect on severe diarrheal disease should also be examined and results should be discussed.

4. It is not clear what is the difference between fig. 1A and 1B. There is no specific mention of fig 1A and 1B in the result section. Please clarify.

5. It is reasonable to assume that if the rotavirus vaccine is effective, the proportion of other enteric pathogens in diarrhea patients should increase. However, the authors need to explain more the assumption in line 16 that higher frequencies of other pathogens in combination with rotavirus infection.

6. The authors should provide summary table and related statistics to show increased other enteric pathogen infection after rotavirus vaccination.

7. In figure 1, the authors showed high infection rates and odds ratios for some of the no-rotavirus enteric pathogens in in rotavirus infected and vaccinated patients. Do authors suggest that rotavirus vaccination increased risk of other enteric infection especially in rotavirus infected patients? Can authors provide hypothesis and references (if any) for the increased rotavirus coinfection with other enteric pathogens after vaccination?

8. The other major objective for the paper is to investigate serotype shit in field rotavirus strains as the result of vaccination pressure. The results were inconclusive. The authors please explain given the duration of the study and the number of samples, what expected of rotavirus serotype changes will be considered as the definite conclusion that rotavirus vaccination significantly alter the serotype of epidemic rotavirus strains?

6. PLOS authors have the option to publish the peer review history of their article (what does this mean?). If published, this will include your full peer review and any attached files.

Reviewer #1: No

Reviewer #2: No

---

## [Author Response · Author response to Decision Letter 0]

15 Mar 2023

Reviewer #1: This is an interesting study with value to add to current literature. The study design is technically sound. My major concern is that causative conclusions are drawn without robust statistical analysis to support them. There is insufficient discussion of the potential limitations of the study, particularly concerning is the absence of discussion of confounding factors that may have contributed to the reduction in frequency of rotavirus infections. Further details on sampling and recruitment methods would also be beneficial - was a sample size calculation performed? Overall, there are some errors in spelling (importantly faces -> faeces/feces) and grammar that should be addressed to make the manuscript clearer.

RE: In the revised manuscript we have avoided statements that might suggest that associations are causative.

As for potential confounders that might contribute to the observed lower number of rotavirus infections, one might consider for example a general increase in the socio-economic status and general hygiene in the society. However, since we did not observe any decline in rotavirus infection rate, mentioning such confounders would be inappropriate.

We have also added more information about sampling and recruitment (page 5, lines 9-12). The samples were collected either as faeces or as rectal swabs. Recruitment to place at study centers which were chosen to allow inclusion of both out-patients (health centers) or in-patients (district hospitals and university hospitals).

Since information about what changes of rotavirus infections that would be expected after the implementation of vaccination was insufficient when the study was planned we did not perform sample size calculation.

We have checked the grammar and spelling.

Reviewer #2: The authors used a case-control study design to indirectly evaluate rotavirus vaccine efficacy in hospital admitted diarrhea patients in Rwanda. They compared rotavirus and other enteric pathogen detection rates in patient samples collected before and after vaccination campaign with Rotataq. The hypothesis is that if the rotavirus vaccine is effective relative proportions of rotavirus infection should be lower and other enteric pathogens should be higher in samples collected after vaccination campaign. The limitation of the paper is that because of the case-control design nature, efficacy was measured as odds ratio while number of previously published studies, including prospective studies in countries of similar economic status demonstrated vaccine protective rates. This study does not offer any significant novel findings. The other objective of the study is to investigate the rotavirus serotype shifts after vaccination with Rotataq. However, given the short period of observation and small sample size, the authors are unlikely to distinguish if the viral serotype changes are the result of vaccination or natural variation. There are other issues the authors need to address.

RE: The study did not have a case-control design, but was a comparison of pathogen detection frequencies in similar pediatric populations included during two periods (before and after the implementation of vaccination), with focus on the comparison between vaccinated and not vaccinated children.

Most of the patients had not been admitted to hospital, but were included at health centers.

We agree that longer observation periods are important for understanding rotavirus serotype changes and to what extent they might be a result of rotavirus vaccination. However, this is a limitation that our study shares with other studies.

1. The authors used pre and post vaccination hospital admitted diarrhea patients to compare the RV infection rates in several age groups. There was not significant protection in age 12-36 month group and the overall odds ratio was moderate in age <12 month (table 2). Their results are consistent with previously reported moderate efficacy of rotavirus vaccine in low income developing countries including African countries. Because of the differences in age groups, the authors should provide more details of Rwanda rotavirus vaccine program. For instance, the vaccination starting age and how many boost immunizations and the vaccination rate in different age groups. The author should discuss if the different vaccine protective efficiency in the two age groups are the results of vaccination schedule or vaccination rate.

RE: The details of the rotavirus vaccine program in Rwanda have been included (page 3, lines 11-12).

2. Table 1, 2, it appears that group of >36 month old had no specific values in the study. Authors should consider to remove this age group from the analysis.

RE: We agree that one might consider to remove this group since it does not contribute to the comparison between vaccinated and unvaccinated. However, we prefer to report the results in line with the original study protocol, and therefore the data for group is still reported, but is excluded from some of the comparisons.

3. Table S1, authors suggested that the rotavirus vaccine worked better to prevent severe dehydrating diarrhea. Since it is one of their major conclusions, the table S1 should be included in main text. Odds ratios for severe diarrhea in rotavirus positive and negative groups should be separately calculated and age difference in vaccine effect on severe diarrheal disease should also be examined and results should be discussed.

This table has been included in the main text, and complemented as suggested.

4. It is not clear what is the difference between fig. 1A and 1B. There is no specific mention of fig 1A and 1B in the result section. Please clarify.

RE: As described in the figure legend on page 10, lines 10-13, these figures show the detection rates and odds ratios (pathogen detection rate in vaccinated/pathogen detection rate in unvaccinated) of other pathogens in children with or without rotavirus vaccination (black = all), including separation of those with (red) of without (blue) detection of rotavirus. Figure 1A shows findings in children below 12 months, B children above 12 months of age. 

5. It is reasonable to assume that if the rotavirus vaccine is effective, the proportion of other enteric pathogens in diarrhea patients should increase. However, the authors need to explain more the assumption in line 16 that higher frequencies of other pathogens in combination with rotavirus infection.

RE: In unvaccinated children rotavirus infection is almost always symptomatic. In vaccinated children it may be asymptomatic. Therefore, vaccinated rotavirus-positive children with diarrhea often have another (causative) pathogen that in our testing will be observed as a co-infection. In rotavirus-negative children one would not expect any impact of vaccination on co-infections. This rationale is described in the Statistics section of Materials and Methods (page 7, lines 15-21).

6. The authors should provide summary table and related statistics to show increased other enteric pathogen infection after rotavirus vaccination.

RE: This is actually shown in the supplementary table that accompanies Figure 1A and 1B.

7. In figure 1, the authors showed high infection rates and odds ratios for some of the no-rotavirus enteric pathogens in in rotavirus infected and vaccinated patients. Do authors suggest that rotavirus vaccination increased risk of other enteric infection especially in rotavirus infected patients? Can authors provide hypothesis and references (if any) for the increased rotavirus coinfection with other enteric pathogens after vaccination?

RE: The proposed explanation is presented in the response above, at comment 5. 

8. The other major objective for the paper is to investigate serotype shit in field rotavirus strains as the result of vaccination pressure. The results were inconclusive. The authors please explain given the duration of the study and the number of samples, what expected of rotavirus serotype changes will be considered as the definite conclusion that rotavirus vaccination significantly alter the serotype of epidemic rotavirus strains?

RE: We agree that the results are inconclusive regarding the cause of serotype changes. We cannot define what serotype changes that can be attributed to rotavirus vaccination. Still, we think it is important to investigate and report the changes of serotype distribution, and as far as we provide novel data on the rotavirus serotypes circulating in Rwanda during the two periods of the study. Over all, our findings of serotype shift over time is in line with what has been reported for other geographic regions.

---

## [Editor Report · Decision Letter 1]

12 Apr 2023

Rotavirus infections and their genotype distribution in Rwanda before and after the introduction of rotavirus vaccination

PONE-D-22-34328R1

Dear Dr. Andersson,

We’re pleased to inform you that your manuscript has been judged scientifically suitable for publication and will be formally accepted for publication once it meets all outstanding technical requirements.

Kind regards,

Mrinmoy Sanyal, PhD

Academic Editor

PLOS ONE

---

## [Editor Report · Acceptance letter]

17 Apr 2023

PONE-D-22-34328R1 

Rotavirus infections and their genotype distribution in Rwanda before and after the introduction of rotavirus vaccination 

Dear Dr. Andersson:

I'm pleased to inform you that your manuscript has been deemed suitable for publication in PLOS ONE. Congratulations! Your manuscript is now with our production department. 

Kind regards, 

on behalf of

Dr. Mrinmoy Sanyal 

Academic Editor

PLOS ONE